# Segment Anything in Medical Images with nnUNet

Raphael Stock[1][0009−0008−5531−1072], Yannick Kirchhoff[1][0000−0001−8124−8435], Maximilian R. Rokuss[1][0009−0004−4560−0760], Ashis Ravindran[1][0000−0003−2942−6235], and Klaus Maier-Hein[1,2][0000−0002−6626−2463]

[1] German Cancer Research Center (DKFZ), Heidelberg, Division of Medical Image Computing, Germany
[2] Pattern Analysis and Learning Group, Heidelberg University Hospital, Germany
`raphael.stock@dkfz-heidelberg.de`

**Abstract.** In this paper, we present an enhanced medical image segmentation approach leveraging the nnUNet framework, specifically tailored to integrate bounding box prompts for improved segmentation accuracy in resource-constrained environments. By incorporating these prompts as binary masks in an additional input channel, we enable more precise and context-aware segmentation. Our methodology employs a 2D slice-wise approach optimized for CPU-based inference through just-in-time (JIT) compiled functions, ensuring efficient processing on standard clinical equipment. Our solution demonstrates robust performance, achieving an average Dice Similarity Coefficient (DSC) of 80.98% and a Normalized Surface Dice (NSD) of 83.23% across multiple modalities in the validation set. This indicates its practical applicability and effectiveness in real-world clinical settings, where computational resources may be limited. By focusing on both accuracy and efficiency, our approach makes advanced segmentation technology accessible to a broader range of healthcare providers, facilitating enhanced clinical decision-making and patient care.

**Keywords:** Medical Image Segmentation · nnUNet · Bounding Box Prompts · CPU Inference

## 1 Introduction

### 1.1 Background and Difficulty of the Challenge

Medical image segmentation is a crucial process in clinical practice, enabling precise quantification of anatomical structures and identification of pathological regions. With the advancement of technology, there is a significant transition in this field from the use of specialized models tailored to specific tasks to the adoption of foundation models that can handle diverse segmentation scenarios. However, this shift is fraught with challenges, particularly due to the variability across different medical domains and the resource limitations faced during inference.

The primary difficulty lies in developing models that can generalize across a wide range of medical image modalities and pathological conditions. Each medical domain, such as radiology, histopathology, and oncology, presents unique imaging characteristics and diagnostic requirements, necessitating highly adaptable models. Furthermore, the computational resources required for inference with state-of-the-art models often exceed the capabilities of standard clinical equipment, particularly in settings where high-end GPUs are not available. This creates a significant barrier to the practical implementation of advanced segmentation tools in many healthcare facilities, thereby limiting their accessibility and utility.

The Segment Anything on Laptop 2024 Challenge is therefore launched to advance the current state of segmentation technologies, aiming to develop truly universal and resource-efficient medical image segmentation models. Recognizing the limitations of existing approaches, the objective of this challenge is to inspire innovations that will result in models deployable on standard clinical equipment, such as laptops or edge devices, without the necessity for GPUs.

To facilitate this, this challenge provides an extensive training dataset comprising over 1,000,000 image-mask pairs, encompassing 10 different medical image modalities and more than 20 types of cancer. This dataset is designed to support the development of models that can generalize across a broad spectrum of medical imaging scenarios. By focusing on lightweight, bounding box-based segmentation techniques, the organizers aim to encourage solutions that not only achieve high accuracy but also maintain efficiency in resource-constrained environments.

The challenge shall drive methodological advancements in the field of medical image segmentation, leading to the creation of universal models with broad applicability. Additionally, by emphasizing ease of interaction and deployment, it is envisioned that sophisticated segmentation tools are more accessible to a wider range of healthcare providers, ultimately enhancing clinical decision-making and patient care.

## 1.2   Related Work and State-of-the-Art Methods

Recent advancements in segmentation models have shown promise in addressing some of the above-mentioned challenges. Models like SAM (Segment Anything Model) [3,5] and its variants, including MedSAM [4], MobileSAM [7], and EfficientViT-SAM [8], represent the state-of-the-art in segmentation technology. SAM and its derivatives have demonstrated remarkable performance in natural image segmentation, leveraging large-scale datasets and powerful computational frameworks.

MedSAM, an adaptation of SAM for medical imaging, has improved performance in medical domains but still requires significant computational resources. MobileSAM and EfficientViT-SAM have attempted to address the resource constraints by optimizing for mobile and edge devices, yet their effectiveness across the wide variety of medical imaging modalities and conditions remains a subject of ongoing research. These models provide a strong foundation but highlight the need for further advancements to achieve the goal of universal, resource-efficient medical image segmentation.

## 1.3   Motivation and Contribution

In approaching this task, we leverage the well-established nnUNet framework [1], renowned for its state-of-the-art performance across various medical imaging tasks and domains. nnUNet's robust out-of-the-box capabilities on new datasets make it a baseline for numerous model developments. However, nnUNet is inherently designed for automatic semantic segmentation of target structures it is trained on, without the flexibility to accept prompts.

Our contribution addresses this limitation by integrating a straightforward, yet effective, method for incorporating bounding box prompts into the nnUNet framework. This enhancement allows nnUNet to adapt to the challenge's requirements for prompt-based segmentation. Additionally, nnUNet's use of a lightweight CNN-based UNet architecture, as opposed to more computationally

demanding transformer-based models, ensures excellent computational efficiency.

Our approach introduces a patch-based processing methodology, which contrasts with conventional methods that process entire images at once. While this patch-based strategy can introduce computational overhead, we mitigate this by optimizing our model for rapid CPU-based inference. We employ just-in-time (JIT) compiled functions to accelerate prediction speed, ensuring our model remains efficient even on resource-constrained devices like laptops and edge devices.

## 2  Method

### 2.1  Using nnUNet as Base Model

For this challenge, we employ nnUNet as our base model due to its proven track record of achieving state-of-the-art performance across various medical imaging tasks. nnUNet's inherent flexibility and robust architecture provide a solid foundation for our modifications, ensuring reliable and high-quality segmentation results. Furthermore, we use the residual encoder blocks (ResEnc) as introduced in [2] for nnUNet.

### 2.2  Incorporating Prompts as an additional input channel

To adapt nnUNet for prompt-based segmentation, we introduce a method of incorporating prompts using channel masks. Specifically, we integrate bounding box prompts as additional input channels, allowing the model to focus on specific regions of interest within the medical images (cf. Figure 1). This approach effectively guides the segmentation process, enhancing the model's ability to accurately delineate target structures based on the provided prompts.

### 2.3  Fully slice wise 2D approach

Given the computational constraints and the need for efficient processing on edge devices, we opt for a full 2D approach. By processing images slice-by-slice rather than in a 3D context, we significantly reduce the computational burden. Although a 2D approach typically sacrifices some performance compared to a 3D setting, it allows us to use a single model across all input image modalities. This unified approach eliminates the need to develop and maintain separate 2D and 3D models, streamlining our contribution and ensuring consistency across different types of medical images. This strategy not only aligns with our goal of achieving resource-efficient segmentation but also simplifies the integration of bounding box prompts into the nnUNet framework.

### 2.4  Training vs. Inference

Our training and inference pipelines are designed to maximize efficiency and performance. For training, all images were preprocessed with z-score normalization of the standard nnUNet framework. During training, we focus on optimizing the model's ability to handle various types of bounding box prompts, using a diverse set of training samples from the provided dataset. A random component of one of the foreground classes of each image is sampled and used for the ground truth mask, as well as to extract a bounding box prompt. This bounding box is augmented by random dilations and concatenated to the input image for training. We employ data augmentation techniques to ensure

the model generalizes well across different medical imaging modalities and conditions (see [1] for details on the pipeline).

For inference, we streamline the process to ensure rapid and accurate segmentation on CPU-based systems. We only predict on a patch, which extends by half the patch-size around the bounding box prompt. Furthermore, just-in-time (JIT) compiling the model architecture minimizes latency and maximizes throughput, ensuring that our approach is viable for real-time clinical applications.

### 2.5   Postprocessing

To enhance the accuracy and usability of the segmentation results, we incorporate a postprocessing step that involves removing all predictions that do not lie in the the region specified by the bounding box prompts. This step refines the segmentation output, removing any extraneous areas outside the region of interest, and ensures that the final results are focused and relevant for clinical interpretation.

### 2.6   Compiling Using OpenVINO

To further optimize our model for deployment on edge devices, we compile it using OpenVINO[3]. OpenVINO is an open-source toolkit for optimizing and deploying deep learning models from cloud to edge. This optimization toolchain converts our trained model into an optimized intermediate representation, enabling efficient execution on Intel CPUs and other compatible hardware. By leveraging OpenVINO, we achieve significant improvements in inference speed and resource utilization (cf. Table 4), making our solution practical for use in real-world clinical settings where computational resources are limited.

In summary, our method combines the strengths of nnUNet with innovative prompt integration, a 2D processing approach, efficient training and inference pipelines, targeted postprocessing, and deployment optimization using OpenVINO. This comprehensive strategy ensures that our model meets the challenge's requirements for universal, resource-efficient medical image segmentation.

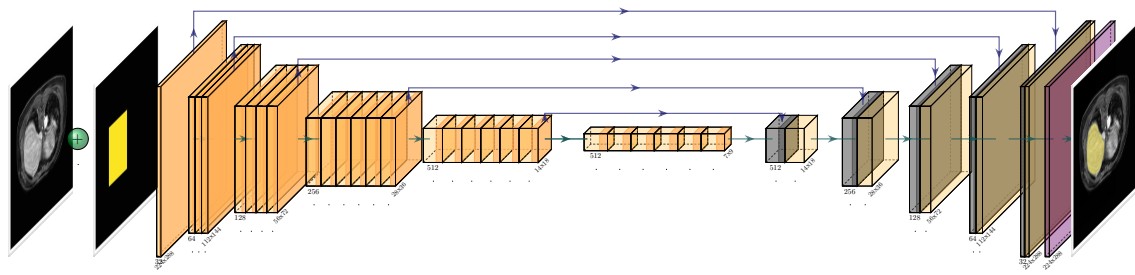

**Fig. 1.** Network architecture: We use nnUNet [1] with residual encoder (ResEnc) blocks [2], which here are illustrated by the orange blocks. The input image is concatenated with the bounding box represented as binary mask and then fed into the network. The patch size of the model is 224x288.

---

[3] https://github.com/openvinotoolkit/openvino and https://docs.openvino.ai/

## 3   Experiments

### 3.1   Dataset and evaluation measures

We exclusively utilized the challenge dataset for model development. The evaluation metrics include two accuracy measures—Dice Similarity Coefficient (DSC) and Normalized Surface Dice (NSD)—alongside one efficiency measure—running time. These metrics collectively contribute to the ranking computation. For model selection we opted for the model showing superior performance across all modalities based on average ranking. We explored the potential of modality-specific models to enhance results for particular modalities. However, we abandoned this "specialist" approach for two reasons. Firstly, selecting the appropriate specialist model would either rely solely on the file name of the case or require an additional modality classification method, which again would increase latency. Secondly, in a real-world setting, switching model weights between predicting images of different modalities would increase latency. We recognize that this switching of model weights is not penalized in the challenge's runtime evaluation, as the docker container is run for each case individually.

### 3.2   Implementation details

**Environment settings**   The development environments and requirements are presented in Table 1.

Table 1. Development environments and requirements.

| System | Ubuntu 20.04 |
|---|---|
| CPU | AMD Ryzen 9 3900X processor |
| RAM | 64GB DDR4-3600 RAM; 256 GB per socket |
| GPU (number and type) | One NVIDIA RTX3090 GPU with 24GB |
| CUDA version | 12.1 |
| Programming language | Python 3.12.2 |
| Deep learning framework | torch 2.2.1 |
| Specific dependencies | |
| Code | |

## 4   Results and discussion

Our proposed method demonstrates robust performance in various medical imaging scenarios, particularly excelling when clear, well-defined anatomical structures are present within the bounded regions. The integration of bounding box prompts effectively guides the segmentation model, allowing it to focus on specific areas of interest and thereby improving accuracy. This method is notably beneficial in common modalities such as MRI and CT, where the target structures often have distinct and recognizable boundaries. In clinical settings, the method proves effective in tasks such as segmenting organs and tissues in high-contrast images, delineating target structures in MRI images, and segmenting regions of interest in X-Ray images with minimal noise or artifacts.

**Table 2.** Training protocols.

| | |
|---|---|
| Pre-trained Model | Not applicable |
| Batch size | 51 |
| Patch size | 224×288×3 |
| Total epochs | 1000 |
| Optimizer | SGD with nesterov momentum ($\mu = 0.99$) |
| Initial learning rate (lr) | 0.01 |
| Lr decay schedule | linear LR decay |
| Training time | 72.5 hours |
| Loss function | Soft Dice loss + Cross Entropy loss |
| Number of model parameters | 71.81M |
| Number of flops | - |
| $CO_2$eq | - |

The method works exceptionally well in scenarios involving the segmentation of organs and tissues in high-contrast images, such as liver and kidney segmentation in CT scans, as well as the delineation of structures such as single teeth or bones in X-Ray images. It is also successful in ultrasound images where the region of interest is well-isolated and less affected by noise or artifacts. In these cases, the bounding box prompts provide a significant advantage by narrowing the focus of the segmentation model, leading to precise and reliable outcomes.

Despite its strengths, the proposed method encounters challenges in certain situations. Primary reasons for failed cases include poor image quality, such as low resolution, significant noise, or artifacts, which can hinder the model's ability to accurately segment target structures. Ambiguous boundaries, particularly in the presence of diffuse pathological regions or overlapping anatomical features, also pose difficulties, as the model may struggle to produce accurate segmentations. Additionally, extreme variability in the appearance of target structures across different patients or imaging conditions can lead to poor generalization by the model.

### 4.1   Quantitative results on validation set

The quantitative results are summarized in Table 3. In addition to presenting the final submitted solution, we conducted ablation experiments to assess the impact of varying training sets on the model's downstream performance. Ablation study 1 involved training our nnUNet architecture solely on 1% of the SAM data, while ablation study 2 employed pre-training on the same 1% subset of SAM followed by training on the provided challenge dataset. Results indicate that utilizing the non-medical dataset of SAM in ablation study 1 yields unsatisfactory outcomes, underscoring the disparity between natural and medical images. Of note, training solely on the provided challenge data appears, on average, to outperform using the pre-trained checkpoint (ablation study 2). However, pre-training showed slight improvements in certain modalities (CT, Dermatology, Fundus). Furthermore, we observed marginal enhancements when pre-training on the entire challenge dataset and subsequently fine-tuning modality-wise on the corresponding subset of the challenge dataset.

**Table 3.** Quantitative evaluation results comparing the performance of the nnUNet architecture under different training conditions. Ablation study 1 involves training exclusively on 1% of the SAM dataset, while ablation study 2 entails pre-training on 1% of the SAM dataset followed by fine-tuning on the challenge dataset. Our proposed and submitted solution was trained solely on the provided challenge dataset from scratch.

| Target | Baseline (LiteMedSAM) | | Ablation Study 1 | | Ablation Study 2 | | Proposed | |
|---|---|---|---|---|---|---|---|---|
| | DSC(%) | NSD(%) | DSC(%) | NSD(%) | DSC(%) | NSD (%) | DSC(%) | NSD (%) |
| CT | 92.26 | 94.90 | 67.64 | 69.14 | 90.60 | 93.37 | 90.05 | 92.84 |
| MR | 89.63 | 93.37 | 43.84 | 44.24 | 81.31 | 85.89 | 82.22 | 86.61 |
| PET | 51.58 | 25.17 | 30.69 | 23.36 | 34.33 | 24.84 | 45.84 | 32.82 |
| US | 94.77 | 96.81 | 64.17 | 68.60 | 80.03 | 86.45 | 80.85 | 87.00 |
| X-Ray | 75.82 | 80.38 | 16.24 | 15.34 | 78.79 | 84.94 | 79.54 | 85.73 |
| Dermotology | 92.47 | 93.86 | 52.95 | 54.28 | 90.29 | 91.84 | 89.75 | 91.36 |
| Endoscopy | 96.04 | 98.11 | 2.41 | 1.55 | 95.07 | 97.31 | 95.75 | 98.06 |
| Fundus | 94.81 | 96.42 | 3.03 | 0.00 | 89.66 | 91.42 | 88.70 | 90.49 |
| Microscopy | 61.63 | 65.39 | 2.26 | 0.43 | 76.06 | 84.38 | 76.11 | 84.13 |
| Average | 83.23 | 82.71 | 31.47 | 30.77 | 79.57 | 82.27 | 80.98 | 83.23 |

Nevertheless, we opted against training modality-specific models due to the aforementioned reasons (cf. section 3.1).

### 4.2    Qualitative results on validation set

In Figures 2 and 3 qualitative results across 10 modalities are shown. While in Figure 2 samples are shown where nnUNet is likely outperformed by LiteMedSAM, Figure 3 shows examples where nnUNet performs well.

### 4.3    Segmentation efficiency results on validation set

Our approach prioritizes efficiency, making it suitable for deployment on edge devices and in resource-constrained environments. By adopting a 2D processing methodology and utilizing just-in-time (JIT) compiled functions, we achieve significant reductions in computational load and inference time, as demonstrated in Table 4. Notably, for CT, MR, Endoscopy, and PET modalities, our method achieves significantly reduced inference times compared to the MedSAM baseline, with the improvement being especially pronounced in ultrasound (US) images.

However, for large 2D images, such as those in Dermoscopy and Microscope imaging, our model experiences a notable increase in inference time over the baseline. This increase is due to our patch-based approach, which necessitates multiple forward passes for large images, resulting in slower inference speeds. This represents one of the major limitations of our method.

Despite this, our model generally demonstrates reduced inference time and lower resource usage, making it capable of running effectively on standard laptops and edge devices without the need for high-end GPUs. This broadens its accessibility and potential impact, allowing advanced medical image segmentation to be more widely adopted in various clinical settings.

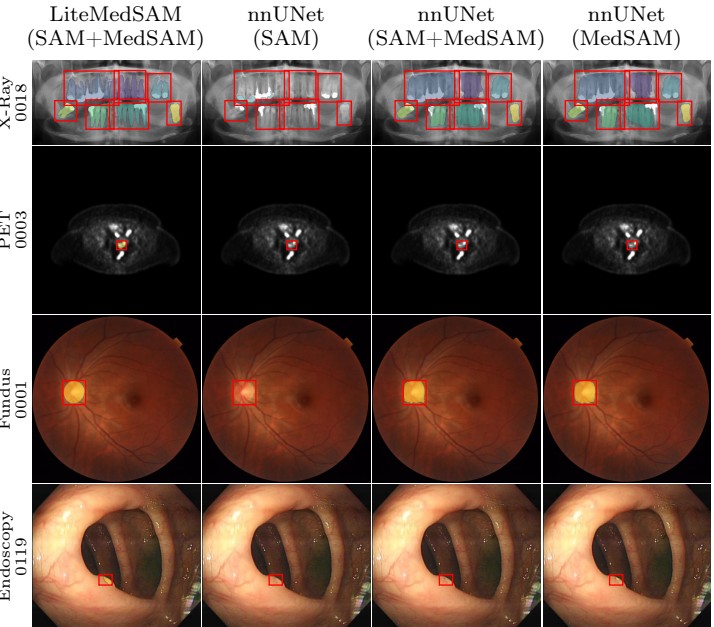

**Fig. 2.** Qualitative results. Within each column the same model was used for prediction. In parenthesis, the training data is indicated, where SAM is the SegmentAnything dataset from Meta and MedSAM the provided challenge dataset. For nnUNet only 1% of the actual SAM dataset was used. The right most model was submitted as final solution. In the first row, nnUNet struggles to understand the box prompt correctly as the dental field is continuously segmented instead of the single teeth within the bounding box. Here, LiteMedSAM correctly understands the intention of the prompt. Despite the different model, this could be due to the larger pre-training on the whole SAM dataset, which could help the model understand the intention of the prompt. The other rows likely also show worse performances of nnUNet compared to LiteMedSAM.

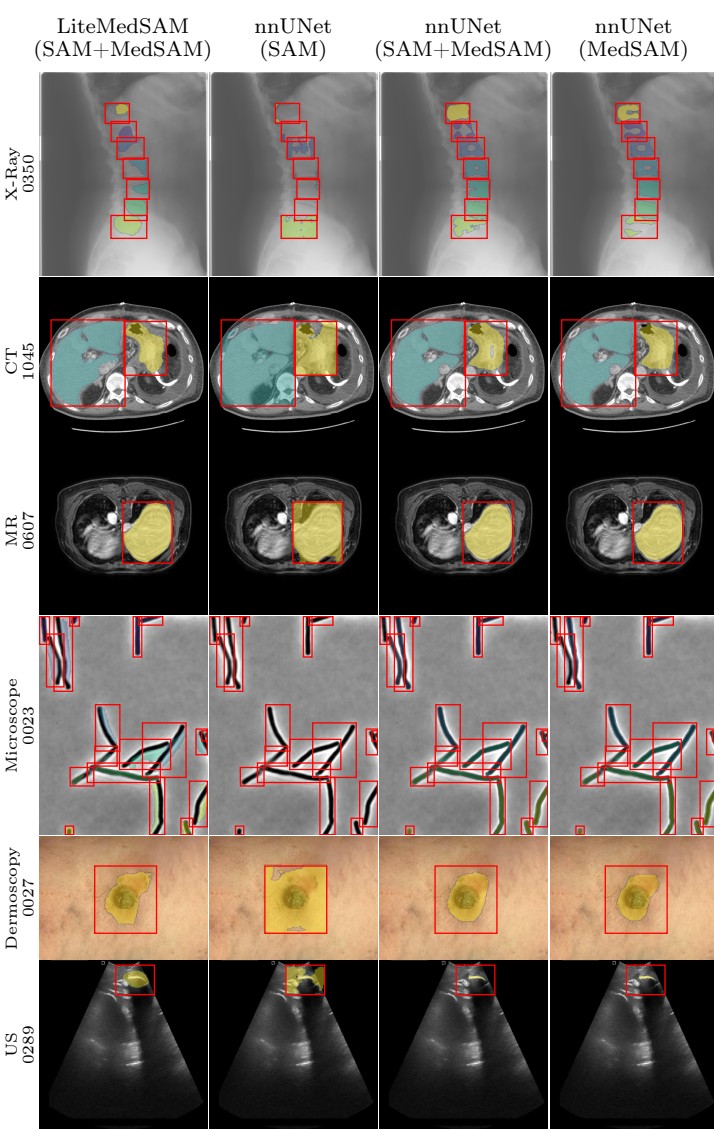

**Fig. 3.** Qualitative results. While under-performing for teeth in the X-Ray modality, nnUNet does a better job at segmenting the vertebrae in X-Rays compared to LiteMedSAM. Furthermore, nnUNet provides seemingly better segmentations for the depicted microscopy image.

**Table 4.** Quantitative evaluation of segmentation efficiency measured in terms of running time (seconds). The baseline is the LiteMedSAM method developed by the challenge organizers. Ablation study presents our proposed solution without OpenVINO optimization, while Proposed refers to the same method post-runtime optimization using OpenVINO. Note: Our reported times for "Ablation Study" and "Proposed" reflect only the time for inference without the model loading and initialization overhead. These times were measured on the machine specified in Table 4.

| Case ID | Size | Num. Objects | Baseline | Ablation Study | Proposed |
|---|---|---|---|---|---|
| 3DBox_CT_0566 | (287, 512, 512) | 6 | 376.4 | 466.70 | 159.03 |
| 3DBox_CT_0888 | (237, 512, 512) | 6 | 100.5 | 97.27 | 35.88 |
| 3DBox_CT_0860 | (246, 512, 512) | 1 | 17.7 | 20.70 | 7.65 |
| 3DBox_MR_0621 | (115, 400, 400) | 6 | 157.1 | 130.39 | 51.48 |
| 3DBox_MR_0121 | (64, 290, 320) | 6 | 99.9 | 83.72 | 31.78 |
| 3DBox_MR_0179 | (84, 512, 512) | 1 | 17.1 | 10.85 | 4.87 |
| 3DBox_PET_0001 | (264, 200, 200) | 1 | 12.1 | 6.93 | 2.75 |
| 2DBox_US_0525 | (256, 256, 3) | 1 | 6.3 | 1.04 | 0.45 |
| 2DBox_X-Ray_0053 | (320, 640, 3) | 34 | 7.3 | 20.52 | 8.28 |
| 2DBox_Dermoscopy_0003 | (3024, 4032, 3) | 1 | 6.5 | 192.03 | 49.10 |
| 2DBox_Endoscopy_0086 | (480, 560, 3) | 1 | 6.1 | 2.99 | 0.89 |
| 2DBox_Fundus_0003 | (2048, 2048, 3) | 1 | 6.1 | 6.15 | 1.87 |
| 2DBox_Microscope_0008 | (1536, 2040, 3) | 19 | 6.8 | 47.41 | 13.97 |
| 2DBox_Microscope_0016 | (1920, 2560, 3) | 241 | 19.1 | 506.83 | 153.71 |
| Total | | 325 | 839 | 1593.53 | 521.71 |

### 4.4 Results on final testing set

This is a placeholder. We will announce the testing results during CVPR (6.17-18)

### 4.5 Limitation and future work

Looking forward, several avenues for further enhancement and exploration are evident. Improving the model's robustness to handle a wider range of image qualities and boundary ambiguities will be a key focus, with techniques such as advanced data augmentation and semi-supervised learning potentially enhancing generalization. While our current approach is 2D-based, integrating 3D context in a hybrid model could combine the strengths of both approaches, offering better performance for complex cases without sacrificing efficiency. Furthermore, addressing the main limitation of increased inference time for large-sized images is a key aspect of future investigation. Potential solutions include optimizing the patch-based system or eliminating the need for patches altogether. Extending the model to handle additional medical imaging modalities and pathologies, including more rare and complex conditions, will further demonstrate its versatility and utility in diverse clinical scenarios.

## 5   Conclusion

In conclusion, while the proposed method shows promise in many areas of medical image segmentation, ongoing refinement and adaptation will be essential to fully realize its potential and address

the remaining challenges. The focus on efficiency and universal applicability positions our approach as a valuable tool in advancing medical imaging technologies and improving patient care outcomes.

**Acknowledgements** We thank all the data owners for making the medical images publicly available and CodaLab [6] for hosting the challenge platform.
This work was supported by the Helmholtz Association's Initiative and Networking Fund on the HAICORE@FZJ partition.

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

**Table 5.** Checklist Table. Please fill out this checklist table in the answer column.

| Requirements | Answer |
| --- | --- |
| A meaningful title | Yes |
| The number of authors ($\leq 6$) | 5 |
| Author affiliations and ORCID | Yes |
| Corresponding author email is presented | Yes |
| Validation scores are presented in the abstract | Yes |
| Introduction includes at least three parts: background, related work, and motivation | Yes |
| A pipeline/network figure is provided | Figure 1 |
| Pre-processing | Page 3 (in section 2.4) |
| Strategies to data augmentation | Page 3 (in section 2.4) |
| Strategies to improve model inference | Page 4 (in section 2.6) |
| Post-processing | Page 4 (in section 2.5) |
| Environment setting table is provided | Table 1 |
| Training protocol table is provided | Table 2 |
| Ablation study | Page 6 (in section 4.1) |
| Efficiency evaluation results are provided | Table 4 |
| Visualized segmentation example is provided | Figure 3 & 2 |
| Limitation and future work are presented | Yes |
| Reference format is consistent. | Yes |
| Main text $>= 8$ pages (not include references and appendix) | Yes |