# OpenReview forum: "Segment Anything in Medical Images with nnUNet"
_thecvf.com/CVPR/2024/Workshop/MedSAMonLaptop — CVPR24 MedSAMonLaptop_

### Official Review · Reviewer_bkrT · 2024-06-12
**The paper presents a well-documented extension of nnUNet to a bounding-box based model that utilized patch-wise inference and just-in-time compilation to improve the efficiency.**

**Rating:** 7
**Confidence:** 4

**Review:**

The authors extend nnUNet to a bounding box-based model that concatenates a binary mask of the bounding box into an additional channel to the model. They fine-tune nnUNet using this strategy on the SA-1B dataset (natural images) and MedSAM's provided training data, stemming from 11 medical imaging modalities.

The paper is written in a clear and easy-to-follow way, and I did not find any typos or formatting mistakes. However, I believe that the paper is not in the challenge template since the padding on the left and right sides is different from the rest of the papers I had to review. I would advise the authors and meta-reviewers to check this for the final camera-ready submission.

The paper is complete and presents all the needed details to reproduce the results, especially since nnUNet is widely used and requires little explanation to run such a pipeline. However, I am not familiar with the concept of patch-wise inference and the authors do not provide enough detail to understand how this actually works. How large is each patch? The authors state on page 4 that "We only predict on a patch, which extends by half the patch-size around the bounding box prompt". However, this is not clear enough, and could easily be elaborated in the form of an additional equation in the camera-ready paper.

For better completeness, the authors can also estimate the FLOPs and CO2 eq (for a single epoch and extrapolate for the whole training) and add them to Table 2 so that it fits the same details as the rest of the papers in the challenge paper track.

Overall recommendation:
I think that the paper has enough details to reproduce the training and evaluation results and lacks only small things that could be easily added in the camera-ready version. Hence, I vote for an acceptance of the manuscript.

Small comments:
Fig. 2: The PET example is barely visible. I would crop to the non-black part of the image and zoom-in so it can be seen what the predictions of the models actually are.

---

### Official Review · Reviewer_Z3La · 2024-06-14
**Systemizing nnUNet for clinical use**

**Rating:** 7
**Confidence:** 3

**Review:**

It’s interesting that you used nnUNet to achieve such good accuracy.
However, there are a few things that I would like to see improved:

1. Standardize the format according to the template provided by the Challenge.
2. The section '2.4 Training vs. Inference' seems to lack details on the strategy for training the model.
3. The details on the fine-tuning dataset and model are insufficient.
4. Providing sources for the various datasets used in the challenge can greatly improve the clarity and reproducibility of your research.

---

### Official Review · Reviewer_yemV · 2024-06-14
**Segment Anything in Medical Images with nnUNet**

**Rating:** 7
**Confidence:** 4

**Review:**

Paper Summary:
The proposed work addresses the Segment Anything In Medical Images on a Laptop challenge via an adapted nnUNet-based approach that incorporates the possibility of adding bounding box prompts. The authors propose using a binary channel as additional input to the model to help the model focus on the region of interest within the bounding box. Their strategy yields competitive results in most of the regarded image domains in the validation set and outperforms the baseline in the microscopy domain by a large margin.

Strengths of the paper:
- Building on top of a well-established baseline is a compelling idea
- The proposed idea of incorporating guidance signals via an additional channel is established and motivated.
- The authors use various engineering approaches to increase the inference speed of the model, such as JIT or Openvino, which are valuable contributions.
- The authors explore the effect of pretraining on natural images, which is interesting.

Weaknesses of the paper:
- While the nnUnet does excel in many different domains (e.g. CT/MRI) in which the network does produce convincing results as the authors have shown, its out-of-the-box performance which involves various preprocessing steps tailored towards the image characteristics of e.g. CT may be suboptimal for domains such as PET. It could be worth investigating this further or stating if certain modifications to the nnUnet pipeline have been made to address this.
- The patch-based approach is not explained clearly. The authors point out that it increases the latency and propose some strategies to mitigate these effects, but the reason why it is useful to opt for a patch-based strategy is not explained.
- While the results are competitive, the approach seems to perform slightly worse than the MedSam baseline on most domains in the validation set

Further ideas
- While the motivation to go for a single model instead of set of expert models can be justified it raises questions about the performance difference. This would be in particular interesting to see if knowledge from one domain is useful to perform better in another domain. This has been ablated from the natural to medical images, but not among different medical imaging modalities.
- The way to incorporate the bounding box prompt could be ablated further (e.g. binary mask vs. distance transforms). A natural question to ask is, how would the approach perform if only the bounding box itself is fed into the model instead of ignoring predictions outside of the bounding box in a post-processing approach?

---

### Official Review · Reviewer_fSFP · 2024-06-16
**Efficient nnUNet segmentation for resource-limited clinical environments**

**Rating:** 8
**Confidence:** 4

**Review:**

The paper's approach of integrating bounding box prompts into the nnUNet architecture for enhanced segmentation in resource-constrained environments is a creative application of existing technologies, suggesting moderate innovation. Employing JIT compilation and OpenVINO optimization for deployment on edge devices indicates a strong technical foundation. The methodological enhancements, including the use of channel masks and the adaptation to a 2D slice-wise approach, are well-justified and align with the goals of efficient processing. The paper presents a well-rounded approach with practical applications and notable enhancements to a well-established segmentation framework. The focus on resource efficiency and real-world applicability, particularly for clinical settings with limited computational capabilities, is commendable.

---

### Decision · Program_Chairs · 2024-10-01

Accept